# Turning Inside Out: Filamentous Fungal Secretion and Its Applications in Biotechnology, Agriculture, and the Clinic

**DOI:** 10.3390/jof7070535

**Published:** 2021-07-02

**Authors:** Timothy C. Cairns, Xiaomei Zheng, Ping Zheng, Jibin Sun, Vera Meyer

**Affiliations:** 1Chair of Applied and Molecular Microbiology, Institute of Biotechnology, Technische Universität Berlin, Straße des 17. Juni 135, 10623 Berlin, Germany; 2Tianjin Institute of Industrial Biotechnology, Chinese Academy of Sciences, Tianjin 300308, China; zheng_xm@tib.cas.cn (X.Z.); zheng_p@tib.cas.cn (P.Z.); sun_jb@tib.cas.cn (J.S.); 3Key Laboratory of Systems Microbial Biotechnology, Chinese Academy of Sciences, Tianjin 300308, China

**Keywords:** filamentous fungi, classical secretion, non-classical secretion, symbiosis, extracellular vesicles, citric acid, glucoamylase, secondary metabolite, pathogen

## Abstract

Filamentous fungi are found in virtually every marine and terrestrial habitat. Vital to this success is their ability to secrete a diverse range of molecules, including hydrolytic enzymes, organic acids, and small molecular weight natural products. Industrial biotechnologists have successfully harnessed and re-engineered the secretory capacity of dozens of filamentous fungal species to make a diverse portfolio of useful molecules. The study of fungal secretion outside fermenters, e.g., during host infection or in mixed microbial communities, has also led to the development of novel and emerging technological breakthroughs, ranging from ultra-sensitive biosensors of fungal disease to the efficient bioremediation of polluted environments. In this review, we consider filamentous fungal secretion across multiple disciplinary boundaries (e.g., white, green, and red biotechnology) and product classes (protein, organic acid, and secondary metabolite). We summarize the mechanistic understanding for how various molecules are secreted and present numerous applications for extracellular products. Additionally, we discuss how the control of secretory pathways and the polar growth of filamentous hyphae can be utilized in diverse settings, including industrial biotechnology, agriculture, and the clinic.

## 1. Fungal Secretion: The Global Bioeconomy and beyond

Filamentous fungi inhabit virtually all marine and terrestrial environments as free living microbes, symbionts, commensals, and pathogens [1,2]. This global colonization is dependent on their ability to secrete a diverse repertoire of protein, organic acid, and secondary metabolite molecules, which enable nutrient acquisition, growth of polar cells termed hyphae, and a number of specialized, niche-dependent functions, ranging from host tissue invasion, symbiosis, sexual reproduction, or killing competing microbes.

Industrial biotechnologists have repurposed filamentous fungal secretion for over a century, where cheap, readily available renewable biomass has been converted to megatons of useful products including hydrolytic enzymes, platform chemicals, and pharmaceuticals [3,4,5]. This efficient recycling of biomass has recently been identified as an important component in the global transition from a petroleum-based economy to a sustainable circular economy, which may ultimately mitigate climate change, achieve international food security, and protect natural ecosystems (Figure 1 [6,7]).

Understanding and (re)engineering filamentous fungal secretion is not, however, only relevant for sustainable applications in industrial biotechnology. As the diversity of their secreted repertoire in the natural niche is explored (Figure 2 and Table 1), a wide range of novel applications have been postulated, from bioremediation of plastic waste [8], using sex pheromones in biosensors [9], to supplying food and medical products during ultra-long space flight [10]. Understanding secretion has also become a major avenue of research in both medical and agricultural biotechnology. Fungi infect a vast range of animals, plants and other organisms, causing an estimated 1.5–2 million human deaths per year, major extinction events, and loss of annual crops sufficient to feed several hundred million people [11,12,13]. Secretion plays multiple and essential roles during human and plant disease, including acquisition of nutrients from the host and delivery of virulence factors. Thus, chemically impeding secretion with novel antifungals, or enabling the effective immune detection of extracellular molecules, promises to reduce both the clinical and environmental impacts of fungal disease.

In order to reflect the interdisciplinary nature of filamentous fungal secretion (Table 1), we ask ‘why do fungi secrete?’ We discuss several distinct, yet interconnected answers, including (i) polar growth, (ii) degradation of macromolecules for nutrient acquisition, (iii) micronutrient uptake, (iv) removal of overflow metabolites, (v) infection, (vi) interspecies communication, (vii) intraspecies communication, and (viii) surviving in the community. We highlight various metabolic and secretory pathways for proteins (Figure 3), secondary metabolites (Figure 4), and acids (Figure 5) and provide some key examples of their applications. Key secretion mechanisms are mainly summarized from model organisms (predominantly Ascomycota). Although there are certainly important differences in secretion machinery across the fungal kingdom (e.g., between Dikarya (Ascomycota and Basidiomycota) and non-Dikarya, between flagellated and non-flagellated fungi, or between phyla [35]) these are outside the scope of this review. For clarity, secretion is summarized in separate sections. However, in the dynamic environments in which filamentous fungi proliferate and colonize, a single cell may be secreting an arsenal of molecules for all the reasons outlined above. Similarly, a separate and fascinating aspect of filamentous fungal secretion is that a single extracellular molecule may play multiple roles connecting, for example, nutrient acquisition and virulence [27], or polar growth and self-communication [22]. In some instances, conventional assumptions to explain why a fungus secretes a certain product have recently been challenged, leading to new avenues of fundamental research and applications [22,36,37]. We begin by discussing the crucial role of secretion in a defining feature of the filamentous lifestyle: polar growth.

## 2. Polar Growth

While spore dispersal can occur across very large distances (including between continents [38]), fungi are in the vast majority of cases non-motile, and consequently filamentous species colonize substrates by extension of highly polar cells termed hyphae. The shape of the hypha is dependent on the cell wall, which is a complex, yet spatially organized, mesh of polysaccharides and proteins (recently reviewed in [39]). Abundant sugars in the cell walls of Ascomycota are 1,3-β-glucan, 1,4-β-glucan, 1,6-β-glucan, and chitin, the latter of which is composed of N-acetylglucosamine monomeric units [40]. Additional sugars include galactomannan, galactosamine, glycosaminoglycans, alpha-1,3 glucans, and others. Although cell wall composition is drastically different in other phyla (e.g., Mucorales have lower levels of 1,3-glucan and higher chitin content compared to ascomycete *Neurospora crassa* [41,42]), the function of the cell wall remains comparable, with cell shape and substrate colonization largely dependent on this structure. Consequently, the various life-styles and applications of filamentous growth is dependent on the cell wall, including branching mycelial networks in terrestrial and marine habitats [43], physical penetration of the host tissue and immune recognition during disease (Section 5), or development of complex macromorphological structures during submerged fermentation (Section 2.5). Secretion plays a crucial role in establishing and maintaining filamentous morphology by delivery of membrane and cell wall synthesizing enzymes to the apex. Recent insights into the coupling of secretion and polar growth, mainly summarized from ascomycete model organisms, are summarized below.

### 2.1. From Dormancy to Growth: Establishing Polarity

A fundamental stage in the filamentous lifecycle is germination, where metabolically dormant spores activate metabolism and begin growth. Favorable conditions for germination of most fungi includes sensing of external carbon and nitrogen nutrients, which may be transduced by cyclic AMP/protein kinase A (cAMP/PKA), mitogen activated protein kinase (MAPK), and/or RAS signaling pathways. The precise molecular and sub-cellular events which occur is not comprehensively understood for most filamentous fungi, with much current knowledge gleaned from the model systems *Aspergillus nidulans* and *N. crassa* [44]. In general, germination involves the entry of water into the spore and a period of isotropic swelling. This swelling also is presumed to cause changes in the cell wall which may be transduced by the above signaling pathways. Next, there is a rapid onset of respiration and protein/RNA synthesis and utilization of stored energy reserves, which may include trehalose, glycogen, or lipids. Subsequently, DNA syntheses commences [44].

The next phase involves establishment of polarity, for which Rho-GTPases are crucial, both in the budding yeast *Saccharomyces cerevisiae* (Cdc24) and filamentous moulds (for example RacA and the Cdc24 orthologue CftA in *Aspergillus niger* [45]). These GTPases localize to a specific site of the plasma membrane, and activate a range of other proteins which recruit and/or organize actin filaments and septins, thus directing polar growth [46,47]. Cell wall growth at the site of polarity produces a nascent germ tube, which continually extends at the apex to form a mature hypha. The precise mechanisms by which cell wall synthesizing enzymes and materials arrive at the site of polarity during very early germination (i.e., immediately pre-germ tube) has not been comprehensively determined. However, the observation of a key endomembrane component of the classical secretory system, the Spitzenkörper (SPK), in some germinating spores [48] suggests it may be via this route. It is, however, certain that the classical secretion pathway is necessary for the polar growth and cell wall synthesis of actively growing, mature hyphae.

### 2.2. The Classical Secretion Route and Hyphal Polarity

In the classical secretion route, proteins produced in endoplasmic reticulum (ER) are loaded into secretory vesicles at the Golgi. These include extracellular enzymes for nutrient acquisition (see Section 3), and, additionally, enzymes for cell wall synthesis or maintenance, including chitin, glucan, and other exopolysaccharide synthases [49,50]. Short and long range transport of loaded vesicles respectively occurs via microtubules and actin filaments via kinesin and myosin motors to the newly established polarity site (i.e., germ tubes) or the mature hyphal apex [49,51,52,53,54]. These vesicles collect at the SPK, which in actively growing hyphae is maintained at a precise sub-apical location. The integrity of the SPK requires various polarisome proteins (for example SpaA, which controls actin cytoskeletal organization at the SPK in *A. niger* [55]). These polarisome proteins are also delivered to the tip via actin; recent work in *N. crassa* demonstrated that two multiprotein scaffold complexes (one of which contained the polarisome protein Spa-2), are transported to the polar tip along actin motors via the adaptor SPZ-1 [56]. Spatiotemporal docking and tethering of SPK located vesicles to the plasma membrane is controlled by a conserved multiprotein complex called the exocyst (which consists of Sec3, Sec5, Sec6, Sec8, Sec10, Sec15, Exo70, and Exo84). Finally, fusion of plasma and vesicle membrane occurs via SNAREs (soluble NSF attachment protein receptors), which releases cargo, including cell wall biosynthetic enzymes, to the plasma membrane of a nascent germ tube or growing hyphae. Genetic or biochemical disturbance of Rho-GTPases, microtubules, actin, SPK, polarisome, exocyst, or SNARE function causes defective polar growth in industrial, disease causing, or model fungal organisms, thus firmly establishing secretion as a crucial component of the hyphal growth mode [50,57,58,59,60,61].

### 2.3. Endocytosis and Branching

Throughout hyphal extension, the continued delivery of growth materials and enzymes to the tip occurs via the classical route. In some fungi, notably *Aspergillus* spp. and other ascomycetes, this process has distinct temporal phases [53] with pulses of influx of Ca^2+^ coordinating sequential rounds of actin polymerisation, exocytosis, and tip extension (reviewed in [62,63,64]). In the vesicle supply centre growth model [65], addition to the plasma membrane by vesicle exocytosis is balanced by endocytosis to maintain polarity. Sub-cellular localisation experiments and mathematical modelling in *A. niger*, *A. nidulans*, and the dimorphic yeast *Candida albicans* suggests that this may occur at a putative subapical actin ring maintained only a few (1–3) microns behind the growing hyphal tip [66,67,68]. The vital role that endocytosis plays in polar growth is supported by titratable expression of putative endocytic components in *A. niger*, which caused branching defects, tip swelling, and even hyphal rupture when expressed at low levels [69].

As growth continues, both apical (tip) and lateral branching can occur, ultimately resulting in the formation of a mycelial network. While this process is extremely diverse between different species and phyla, it is thought to be broadly analogous to spore germination (i.e., by polarity establishment and maintenance, recently reviewed in [70]). Interestingly, it is hypothesized that apical branching may be necessitated by excessive delivery of secretory vesicles to the tip [70]. At least in some fungi, lateral branching at intercalary hyphal regions requires the formation of a new SPK [70], thus further supporting the role of classical secretion in branch formation.

### 2.4. Repurposing Polar Growth and Classical Secretion in Biotechnology: A Case Study Using A. niger RacA and Glucoamylase

As a defining feature of filamentous fungi, the coupling of polar growth and secretion at the hyphal tip obviously has intrinsic value to many applications involving individual cells or mycelial networks. However, biotechnologists have begun to specifically (re)engineer polar growth and/or secretion to maximize titres of extracellular products. One recent example used the industrial organic acid and protein producer *A. niger* to enhance secretion a desired enzyme by up to 400% [15]. This study focused on the glucoamylase GlaA, a major *A. niger* secreted protein which is used to saccharify starch to glucose in a variety of food industries with a global value of over $1 billion/year [15,71]. Prior to this study, deletion of the Rho-GTPase encoding gene *racA* in *A. niger* had been demonstrated to result in an actin localization defect, loss of polarity at the tip, and a hyperbranching phenotype during submerged growth [47]. Given that most protein secretion occurs at the apex, it was hypothesized that the *racA* mutant might be a useful chassis strain for protein secretion due to elevated tip numbers in this mutant (~20% increase) and comparable biomass relative to the progenitor control [15]. By placing the *glaA* gene under control of a tetracycline inducible promoter (tet-on), expression of this gene was elevated above native transcriptional levels based on addition of doxycycline to growth media [72]. In order to track post-Golgi vesicles in this mutant, the v-SNARE SncA was labelled with GFP. Finally, *racA* was deleted to generate a hyperbranching chassis [15]. Intriguingly, synthetically elevated expression of *glaA* was compensated for by an increase in post-Golgi vesicles as determined by quantitative microscopy of SncA. Moreover, by elevating *glaA* expression levels in the *racA* mutant, four times as much protein was detected in growth medium compared to controls [15]. Thus, the fungal secretory system is able to accommodate user-defined increases in secretory proteins by generating more vesicles, and, additionally, product titres can be increased in a *racA* hyperbranching mutant. This study has led to similar emerging industrial applications using *racA* mutants with defective polar growth and elevated branching, e.g., to increase cellulase production by *Trichoderma reesei* [57].

### 2.5. Controlling Filamentous Growth for Optimal Macromorphologies

In liquid bioreactors used for fermentation of proteins, organic acids, and secondary metabolites, filamentous fungal growth results in the formation of various macromorphologies, including loosely dispersed hyphal fragments, aggregated clumps, and spherical pellets several millimetres in diameter [3,73]. From a process engineering perspective, each macromorphology has crucial implications for performance of the bioreactor and product titres. Dispersed growth results in high levels of nutrient and oxygen transfer into the cell, but, conversely, causes high medium viscosity and thus nutrient and oxygen gradients in the fermenter. Pelleted growth, in contrast, results in low viscosity, and is associated with an absence of nutrient gradients, reduced energy input to stir the bioreactor, and an elevated resistance to sheer stress [74]. However, pellets may have low nutrient transfer into the centre of the cellular mass, thus potentially limiting productivity [75].

As described for fungal *racA* mutants above, modifying fungal morphology for a maximum tip:biomass ratio may be a successful strategy for improving protein secretion [15,76,77]. Various other correlations between productivity and macromorphology have also been described for organic acids and secondary metabolites ([78,79,80,81] reviewed in [3]).

Submerged macromorphology can be controlled to a limited extent by modifying a range of abiotic growth parameters, such as concentration of spore inoculum, temperature, stir speed, pH, carbon/nitrogen sources, magnesium ion availability, addition of microparticles to growth media, and others (reviewed in [3]). In general, these parameters are thought to impact the degree of spore/hyphal aggregation or fragmentation, which results in formation of dispersed or pelleted morphologies respectively [73]. There is, additionally, growing evidence to suggest that in future, user-defined macromorphological growth parameters can be achieved by genetic control of filamentous growth morphologies. In *A. niger*, for example, tet-on controlled expression of a putative endocytic component *aplD* was used in gene titration experiments to generate hyphae of various lengths and branch phenotypes, which could be quantitatively correlated with various macromorphological features, such as pellet formation, diameter, and aspect ratio [69]. A recent methodological breakthrough has been the quantification of hyphal growth in the pellet interior using X-ray microtomography techniques (micro-CT, [82]). This technology revealed highly complex branching phenotypes within the pellet core for multiple industrially used fungi [82], enabling the quantitative estimation of nutrient diffusivity into the interior [75], and opening up new avenues for controlling productivity by engineering polar growth inside fungal pellets.

### 2.6. New Additions to the Classical Secretion Pathway: Extracellular Secretory Vesicles

An exciting discovery in the field of fungal secretion that may also have major implications for white biotechnology was the production of extracellular vesicles (EVs) by *Cryptococcus neoformans* in 2008 [83,84]. Since then, EVs have been identified in model budding yeast (*S. cerevisiae*, [85,86]), the causative agents of human or plant diseases, and others (*Paracoccidioides brasiliensis*, *C. albicans*, *Histoplasma capsulatum, Fusarium oxysporum*, *Zymoseptoria tritici*, [87,88,89,90]). EVs are heterogenous in size, e.g., ranging from 50 and 250 nm in diameter from *Z. tritici*, cross the cell wall into the extracellular space, and can contain a variety of cargo, including enzymes [85,88], RNA [91], lipids [92], and polysaccharides [93].

Work in *S. cerevisiae* indicates that components of the classical secretory route play a role in EV formation or loading, as deletions in various regulatory proteins, such as the Rab-GTPase Sec4, resulted in modified EV protein cargo [86]. Similarly, yeast Golgi mutant isolate display defective EV release kinetics from the cell [86]. More recently, the yeast endosomal sorting complex required for transport (ESCRT) has also been indicated in EV formation [85]. As a component of the fungal endomembrane system, the ESCRT is known to processes endocytosed surface proteins for lysosome or vacuole degradation. By screening EVs from yeast ESCRT mutants, this complex was demonstrated to impact diameter, protein content and abundance of EVs [85]. Thus, it is likely that the vesicle trafficking and exocytic/endocytic machinery necessary for classical secretion may also generate fungal EVs.

Given the very recent discovery of EVs in fungi, the function of these secreted particles and their applications are still emerging. It seems likely that they are associated with fungal growth/homeostasis via cell wall remodeling, as many cargo proteins are involved in cell wall synthesis, including glucan synthase Fks1 and chitin synthase Chs3 in *S. cerevisiae* [85,94]. For human infecting fungi (see Section 6), it is also possible that they are involved in virulence, as both *C. albicans* and *C. neoformans* EVs contain immunomodulatory cargo and activate mammalian immune cells (recently reviewed in [94]). During plant disease, the role of EVs is currently less clear, but may also involve virulence. This hypothesis is supported firstly by the secretion of polyketide biosynthetic enzymes in *F. oxysporum* EVs [88], and secondly by the well-established role of extracellular fungal secondary metabolites as phytotoxic virulence factors (see Section 6). It is worth noting that the cargo is both extensive and unique: from 240 proteins identified in *Z. tritici* EVs, only 30 were also found in the conventional secretome as determined by LC-MS/MS analyses [87]. These data support the notion that EVs are indeed an important secretion route. 

The diverse protein, lipid, and nucleic acid EV cargo which has already been identified open up potentially new avenues for generating hyperproduction isolates in white biotechnology. The discovery of EVs produced by industrial strains is possible, and future comprehensive understanding and engineering of their generation and secretion may enable a convenient route to produce high titres of a desired extracellular molecule.

### 2.7. Summary

Polar extension of hyphal cells underpins substrate colonization by filamentous fungi, and secretion via the classical route is absolutely necessary for this mode of growth. In biotechnology, both secretion and polarity have begun to be controlled using abiotic parameters in fermenters, and by generating mutant strains using conventional and newly developed genome editing technology [95]. Ultimately, this will enable the next generation of highly efficient cell factories with optimal morphologies and maximum product titres to be constructed. Polar growth is also vital for other lifestyles and niches of filamentous fungi, ranging from infection of the host to free living microbial communities. These are discussed in subsequent sections.

## 3. Breakdown of Macromolecules for Nutrient Acquisition

### 3.1. A Stomach Outside the Body: Digesting the Extracellular Space

The remarkable protein secretory capacity of filamentous fungi (e.g., over 30 g/L glucoamylase in *A. niger* and 100 g/L of cellulases in *T. reesei* [16], recently reviewed in [14]) is often directly related to their necessity to degrade the external environment for nutrient uptake. With a few hypothesized exceptions [96,97], fungi are heterotrophic, meaning they are unable to generate their own metabolites and energy from inorganic sources by photosynthesis. Consequently, they are prolific secretors of hydrolytic enzymes, which break down a diverse range of complex extracellular polymers such as lipids, proteins, lignin, and cellulose into simple molecules for osmotrophic absorption of carbon and nitrogen. While some such enzymes are secreted by poorly characterized mechanisms known as the non-classical or unconventional route (e.g., the aspartic protease PepN in *A. niger* [98]), most extracellular hydrolytic enzymes follow the classical secretion pathway (Section 2), thus coupling polar growth and nutrient acquisition. Degradation of and acquisition from the external environment make fungi the largest degraders of biomass in terrestrial ecosystems, prolific recyclers of waste materials, and efficient producers of useful enzymes [5,99]. 

### 3.2. Secreted Lipases in Industrial Biotechnology: An Exemplar Success Story

Secreted enzymes used in industrial biotechnology include but are not limited to lipases, cellulases, glucoamylases, pectinases, phytases, proteases, xylanase, glucose oxidases, and hemicellulases which are isolated from production strains belonging to *Rhizopus, Aspergillus, Trichoderma, Penicillium, Fusarium, Thermothelomyces*, and many other genera. Secreted lipases, as just one example, are enzymes that hydrolyze extracellular triacylglycerols to fatty acids and glycerol [100]. Fermentation media containing glycerol or fats are known to induce lipase secretion, and some industrially utilized fungi able to grow using lipids as the sole carbon source. A large portion of fungal lipases are extracellular: a screen of the online genome compendium FungiDB [101] returned a total of 2383 fungal genes belonging to the gene ontology term Lipase (GO:0016298). Of these, 1015 predicted lipases encoded a putative N-terminal signal peptide, suggesting that over 40% are secreted. Fungal lipases are used in many industries due to their efficient activity over a range of temperatures and pH, and are sold commercially as Lipozyme^®^ (*Thermomyces lanuginosus*, Novozymes), Lipase FE-01 (*Aspergillus oryzae*, ASA Spezialenzyme), among others [17]. These industries include biodiesel (degrading oil feedstocks for production), dairy (hydrolysis of milk/butter fats), textile (fabric treatment), detergent (additive to improve fat removal), paper (removal of pitch), pharmaceutical (hydrolytic/esterification reactions in drug synthesis), and leather (degreasing) [18,102]. Lipases have also been identified as potential solutions to clean oil spills following industrial disasters [103]. The diverse applications of fungal lipases are one example from many which demonstrate that repurposing fungal secretion of hydrolytic enzymes has been extremely successful in biotechnology. 

### 3.3. New Frontiers: Degrading Lignocellulose and Plastics

The efficient use of lignocellulose waste material from agriculture (e.g., straw/corn stover) and forestry (e.g., sawdust) as an industrial feedstock would be a major step towards a sustainable bioeconomy. Components of the plant cell wall includes cellulose (d-glucopyranose linked by β-1,4-glycosidic bonds), hemicellulose (monomeric sugars also linked by β-1,4 and β-1,3-glycosidic bonds) and lignin (cross-linked phenolic polymers with a complex three-dimensional structure [104]). Cellulose is embedded in a lignin matrix, thus shielding this molecule from enzymatic degradation and making the breakdown of lignocellulose a major bottleneck to industrial applications. In terrestrial and marine ecosystems, brown and white rot basidiomycete and other fungi secrete a range of hydrolytic enzymes (which are part of the carbohydrate active enzymes, or CAZy) and possess an oxidative degradation system for lignocellulose breakdown and nutrient acquisition (recently reviewed in [105]). However, these applications are too slow for large scale industrial use, and consequently, engineering lignocellulose degradation from a plethora of fungi is a major objective for current research [6,105]. 

One promising advance has been effective community level mining of fungal genomes for putative CAZy genes, including those encoding putative glycoside hydrolases, glycosyl transferases, polysaccharide lyases, carbohydrate esterases, binders of carbohydrates, and auxiliary activity enzymes (e.g., lytic polysaccharide monooxygenases and other lignin degradation enzymes [106,107]). Encouragingly, specialized in silico tools for identifying CAZy genes in fungal genomes have recently developed [108], which will enable high priority candidate genes to be tested in heterologous expression experiments using genetically tractable industrial production isolates. 

The most prolific producers of cellulase enzymes, for example, are *Thermothelomyces thermophila* and *T. reesei*. For several decades, the publicly available *T. reesei* isolate RUT-C30 has been used to produce 30 g/L cellulases [109], with mutagenized industrial strains able to generate over 100 g/L (reviewed in [16]). More recently, thermostable cellulase producers such as *T. thermophila* are an attractive addition to the biotechnology toolkit, as these organisms can tolerate high temperatures encountered throughout biorefinery processes [110]. 

In addition to these genetically tractable and industrially harnessed fungal cell factories, novel and promising lignocellulose degraders have been recently discovered. For example, genome and secretome analyses of a white rot fungus *Peniophora* sp. CBMAI 1063, which had been isolated from a marine sponge in Brazil [111], identified 310 putative ligninolytic enzymes encoding genes from over 17,000 predicted ORFs. These included 17 laccases, 16 peroxidases, and seven glyoxal/alcohol oxidases, all of which contained putative signal peptides, indicating secretion via the classical route (see Section 2). Secretome analysis from bioreactor cultivations identified 126 proteins, with over 50 putative CAZymes, strongly suggesting that this strain is a promising resource for new lignin degrading enzymes. This hypothesis was confirmed by purification of the laccase Pnh_Lac1, which was able to promote lignin solubilization and depolymerization from sugarcane bagasse as determined by a UV-light absorption assay [111]. Thus, this work demonstrated that bioprospecting from marine organisms is a successful strategy for discovery of novel lignin degrading enzymes for future uses in white biotechnology. Marine fungi also serve as similar reservoirs for novel secondary metabolite discovery, such as bioactive alkaloids (reviewed in [112]).

A recent and exciting development has been the degradation of polyurethane plastic waste by some fungi [113,114]. For example, isolated endophytes from plants in the Amazon, such as *Pestalotiopsis microspora*, were able to utilize polymer polyester polyurethane as the sole carbon source [113]. Biochemical inhibition experiments using of crude supernatant extracts demonstrated that a secreted serine hydrolase-like enzyme was responsible for polyurethane degradation [113]. More recently, other fungi, including *Aspergillus fumigatus*, *Penicillium chrysogenum*, and various *Cladosporium* species, are also able to utilize polyurethane as a carbon source [115]. Therefore, secreted fungal enzymes may soon enable the industrial remediation of post-consumer and post-production plastic waste (recently reviewed in [8]).

## 4. Secretion and Micronutrients

Fungal growth and metabolism require numerous micronutrients, including iron, copper, potassium, calcium, and magnesium, among others. During in vitro cultivation, these molecules can be readily supplied in so-called vitamin and trace element solutions. In the natural niche, however, they may become limited, e.g., due to utilization by competing microbes, or sequestered by the host during infectious growth (see Section 6, Section 7 and Section 8). Several classes of membrane transporters enable the efficient uptake of micronutrients (e.g., high affinity copper transporters [116], or reductive iron transport mechanisms [117]). Nevertheless, under certain nutrient limiting conditions, fungal secretion is necessary for micronutrient uptake. 

### 4.1. Siderophores and Iron Availability

Iron ions are essential due to their role as cofactors in many enzymatic reactions and catalysts in electron transport systems. Ferric and ferrous iron are often abundant in organic debris as oxides or hydroxides, but presumably become limited in growing microbial communities. Certainly, iron is limited to the invading fungus during mammalian infection, thus making acquisition of this micronutrient a vital component of host colonization (see Section 6). One strategy employed to acquire iron when limited is the synthesis iron chelating molecules called siderophores. In the soil decomposer and human pulmonary pathogen *A. fumigatus*, for example, extracellular siderophores fusarinine C and its derivative triacetylfusarinine C are produced from amino acid precursors by nonribosomal peptide synthetases (NRPS) and multiple subsequent enzymatic steps [28,118]. In contrast to secreted hydrolytic enzymes, fungal secondary metabolites are not generated in the ER, but at other subcellular locations. Precursors for extracellular *A. fumigatus* siderophores includes conversion of L-glutamate to L-ornithine in mitochondria, and, additionally, conversion of mevalonate to the intermediate 5-anhydromevalonyl-*N*^5^-hydroxy-l-ornithine in peroxisomes (reviewed in [28]). The final siderophore product is then secreted by an as yet unknown plasma membrane transporter, where they sequester external Fe3+ ions before transport back into the cell by a membrane transporter MirB [118]. 

Thus, non-classical secretion of some secondary metabolites are also an important component of fungal nutrient acquisition in certain limiting conditions (see Section 6, Section 7 and Section 8 for further discussions on the function of extracellular secondary metabolites). It is worth noting that various applications have been postulated for fungal siderophores, including in medicine (i.e., by treating conditions which lead to excess iron), agricultural biocontrol (to limit microbial growth in crop soils), and the decontamination of certain toxic metals during energy production [119,120].

### 4.2. Organic Acids May Liberate and Chelate Nutrients from the External Environment

The majority of industrially relevant organic acids secreted by filamentous fungi are derived from the tricarboxylic acid cycle (TCA), and include citric, oxalic, itaconic, fumaric, malic and succinic acid, among others. In the natural niche, it is possible that reduced pH of the external environment provides a competitive advantage over neutrophilic microbes (see Section 8). For plant degrading fungi, such as *A. niger*, organic acids have been hypothesized to also aid degradation of plant cell walls [121,122]. An alternative explanation is that acidification of the external environment may be an effective mechanism to liberate both macronutrients (such as phosphate in soil [123,124,125]) and micronutrients (such as iron, copper and zinc [122,125]). A recent study which supports a micronutrient acquisition hypothesis of fungal organic acid secretion suggests that citric acid may be produced as a siderophore by *A. niger* [36]. In this work, the citric acid synthase encoding gene, *citA*, was demonstrated to have elevated expression under iron limiting conditions when compared to iron replete controls. Moreover, citric acid secretion was inversely correlated with iron concentration in growth media, an observation which was also independent of nitrogen source. Finally, the addition of Fe(III) citrate was able to rescue growth limitations under iron limiting conditions, suggesting that citric acid may indeed function as an extracellular siderophore under certain nutrient limited conditions [36]. Similar roles of citric acid have also been postulated for some bacteria [126]. Taken together, these data suggest that organic acid secretion by some fungi may be a mechanism to extract micronutrients from the environment, and thus new avenues for maximizing production can be postulated from limiting these ions in growth media. More generally, the many applications of these acids may be a further example of biotechnologists repurposing fungal nutrient acquisition (see Section 5 for an alternate explanation of secreted organic acids as overflow metabolites).

### 4.3. Summary

In addition to polar growth, an essential function of fungal secretion is acquisition of nutrients. From the perspective of useful enzymes, many are repurposed to degrade and recycle complex organic molecules from industrial feed stocks. Additionally, certain secondary metabolites (e.g., NRPS derived siderophores) and some organic acids may also facilitate nutrient uptake into the cell. 

## 5. Removal of Overflow Metabolites 

### 5.1. Organic Acids as Overflow Metabolites

In some instances, fast growth in nutrient rich conditions may generate so called overflow metabolites, whereby growth substrates, such as glucose, are incompletely oxidized. In fungi, this may include the variety of TCA derived organic acids, such as citrate and itaconate [34]. Under the overflow hypothesis, a rapidly growing cell more efficiently uses limited protein and energy resources by limiting the costly biosynthesis and activity of respiratory enzymes. The metabolic trade-off is partial oxidation of a given carbon source, which thus generates mitochondrial build-up of certain TCA intermediates. These molecules are subsequently exported to the cytosol, and then excreted. Another related explanation for organic acid secretion under nutrient-rich conditions is that the rapid uptake of glucose (both by high affinity transporters and diffusion) combined with high glycolytic flux leads to an unfavorable intracellular build-up of TCA intermediates which are subsequently secreted [127]. Therefore, the secretion of organic acids in the nutrient-rich media employed by biotechnologists may be a fungal response to TCA overflow metabolites, rather than (or in addition to) a response to iron or other nutrient limitation [36]. 

### 5.2. Increasing Glycolytic Flux for Organic Acid Hypersecretors

Numerous approaches have been applied in the last few years to increase glycolytic flux for enhanced organic acid production strains (recently reviewed in [32]). One promising update which could be broadly applicable to many industrially used organic acid secreting fungi came from a study on the gene *pyrG* in *A. niger* [128]. Disruption of this gene, which encodes an orotidine-5′-decarboxylase, generates a uridine auxotrophy and thus enables the use of *pyrG* as a transformation marker in mutant isolates [4]. An interesting and unexpected observation during routine generation of *pyrG* mutants was that these strains generated elevated citric acid titres during submerged growth compared to progenitor controls [128]. LC-MS-MS analysis demonstrated that citrate, and its precursors acetyl-CoA and oxaloacetate, significantly increased in the *pyrG* mutants. These data are suggestive that *pyrG* deletion might be a facile and simple approach to enhance glycolytic flux for organic acid secretion, and given that this gene is highly conserved in fungi, this may become employed in various industrial cell factories.

### 5.3. Secretion of Organic Acids and Their Applications

Recent work has comprehensively confirmed the mechanisms which underpin how organic acids are secreted. This occurs independently of the classical secretory route (Section 2) and begins with the transport of an organic acid metabolite across the mitochondrial membrane into the cytosol. Transporters which perform this function include, e.g., the *A. niger* citrate/malate shuttle CtpA [129]. For itaconate production in *Aspergillus terreus* and *Ustilago maydis*, the intermediate cis-aconitate is transported out of the mitochondria by MttA and Mtt1 respectively [130,131]. Once in the cytosol, these molecules can be further metabolized, e.g., cis-aconitate is converted to itaconate by the decarboxylase CadA [130]. Finally, organic acids are exported from the cell by membrane transporters, including CexA (citric acid, *A. niger*), DctA (malic acid, *A. niger*), MfsA (itaconate, *A. terreus*), and Itp1 (itaconate, *U. maydis*) [33,129,130,131,132,133]. In various instances, overexpression of the plasma membrane transporter enables drastically increased product titres in shake flask models of fermentation, suggesting that these genes are promising leads for future development of industrial hypersecretion strains [33,132].

Secreted organic acids are used in a diverse range of applications, including as flavour enhancers in the food and beverage industries, or as antioxidants, preservatives, acidulants, pH-regulators, feedstocks or cleaning products pharmaceutical, cosmetics, or bulk chemical industries [3,4,134]. The citric acid market, for example, was recently estimated to be worth $3.6 billion per year [4]. In summary, one of the most useful class of molecules- TCA derived organic acids- may be secreted due to the fungal necessity to rebalance intracellular metabolite levels. The next section discusses another major explanation for fungal secretion: the infectious lifestyle.

## 6. Infection

Disease causing fungi secrete an arsenal of proteins, acids, and secondary metabolites which can initiate infection, perturb or disrupt host immune responses, cause tissue death, and acquire nutrients from the host (see Section 3 and Section 4), among numerous other functions. Proteins which facilitate infection are secreted via the classical pathway [24,25], in EVs [87], via unconventional routes [26], and, in some plant infecting fungi, via a newly discovered host derived structure termed the biotrophic interface complex (see [24]). Secretion routes for the diverse repertoire of secondary metabolites during infection are less comprehensively understood, but presumably include secretion by plasma membrane transporters (e.g., siderophores [28]), EVs [88], and possibly the vesicle based route (recently reviewed in [29]). Certain organic acids which are necessary for full fungal virulence (e.g., oxalic acid during plant infection [122]) are also exported from the cell by membrane transporters (see Section 5). More recently, the role of secreted polyunsaturated fatty acids (PUFAs) during infection has gained attention in mediating both plant and mammalian infection (reviewed in [135]). While the precise roles of extracellular fungal molecules during host–pathogen interactions and disease are extremely diverse and multifaceted (reviewed in [25,26,29,135,136]), some broad aspects are summarized here, and their applications discussed in the subsequent section.

### 6.1. Secreted Molecules and Immune Evasion during Plant or Human Infection

One role of secreted proteins during infection is to mask the detection of pathogen associated molecular patterns (PAMPs) by host pattern recognition receptors (PRRs). One such PAMP, for example, is chitin. Plants secrete numerous chitinases, which liberate oligomers that subsequently activate host immunity by multiple lysin motif (LysM)-containing cell surface receptors (such as chitin elicitor-binding protein CEBiP in rice [137]). In order to inhibit detection by the host via this mechanism, plant infecting fungi also secrete LysM proteins, for example Slp1 in the causative agent of rice blast disease *Magnaporthe oryzae* [138]. Pathogen LysM effectors efficiently bind chitin and can accumulate between the fungal cell wall and host plasma membrane [138], thereby inhibiting the activation of cognate PRR (see [139] for a recent review). Similar functions of secreted LysM proteins have recently been confirmed during fungal infection of insects [140]. These proteins are part of the extremely diverse fungal ‘effector’ arsenal, whereby many small molecular weight proteins are secreted to manipulate and attack the host (recently reviewed in [25]).

Comparable immune evasion strategies are performed by secreted proteins during human infections. For example, the fungal cell wall PAMP β-1,3-glucan is recognized by the c-type ligand dectin-1 [141]. It was recently demonstrated that the dimorphic yeast *C. albicans* secretes the exoglucanase Xog1 to ‘shave’ pathogen β-1,3-glucans from the surface of the cell, thus hiding the fungus from dectin-1, ultimately facilitating infection [142]. Alternatively, secreted proteases from spores and hyphae are known to degrade and inactivate the mammalian complement system during infection of the pulmonary pathogen *A. fumigatus* [143,144] and also during *C. albicans* infection [145,146]. Similar evasion strategies may also be required for other immune-activating cell wall sugars, such as galactomannans [147] or galactosaminogalactan [148].

### 6.2. Direct Host Attack by Secreted Secondary Metabolites and Proteins

#### 6.2.1. Plants

In addition to hiding fungal pathogens from host immune surveillance, secreted molecules can directly attack the host at various stages of infection. Secreted secondary metabolite phytotoxins, for example, disrupt plasma membrane integrity [149], inhibit ATP hydrolysis [150], perturb actin dynamics [151], trigger programmed cell death [152], and manipulate plant growth [153] (reviewed in [154,155,156]). Secreted effector proteins also perform multiple attacking functions during disease, with many hundred often secreted by a single invading fungus (recently reviewed in [25]). One example of an elaborate strategy involves effector proteins translocating into the host nucleus and modulating gene expression to favor infection. For example, recently two secreted proteins were characterized, from the rice infecting pathogen *M. oryzae* termed Host Transcription Reprogramming 1 and 2 (Htr1/2) [157]. These proteins ultimately bind to target gene promoters in the host, and act as repressors for multiple immunity associated genes. Interestingly, *M. oryzae* harnesses two distinct secretory routes for effectors, with proteins that remain extracellularly secreted via the conventional route, whereas effectors which enter the host cell are preferentially secreted via the biotrophic interphase complex [24].

#### 6.2.2. Humans and Other Mammals

Secreted secondary metabolites may also attack host cells and tissues during human disease. In murine infection models for the pulmonary pathogen *A. fumigatus*, for example, numerous secondary metabolite biosynthetic gene clusters (BGCs) are upregulated 14 h post infection relative to developmentally matched in vitro controls [158]. Additionally, *A. fumigatus* isolates which are deficient in production of the cytotoxic molecule gliotoxin are attenuated during murine infection [159,160,161]. With regards to secreted proteins and attack of human cells, batteries of proteases presumably degrade proteinaceous host tissues [162], although certain proteases and oligopeptide membrane transporters are functionally redundant in some infection models of fungal disease [163]. One recently discovered toxic protein is the 31 amino acid candidalysin which is secreted by *C. albicans* hyphae [164]. Candidalysin is able to form pores on the epithelial cell plasma membrane, leading to host cell lysis and successful transfer from commensal to infectious growth [164].

### 6.3. Applications of Fungal Secretion and Extracellular Molecules

#### 6.3.1. Purified PAMPs in Agriculture

One successful application of secreted or extracellular pathogen molecules has been to elicit crop immunity using purified fungal PAMPs. Purified *S. cerevisiae* cell wall, for example, results in transcriptional activation of immunity in grapevine [165], and over the last decade has been marketed as Cerevisane^®^ for protection of lettuce and other crops against mildews and other fungal pathogens. Similarly, over the last decade, the European Union licensed laminarin (marketed under the name Vacciplant^®^), which is used to protect strawberries, apples, and other crops against powdery mildew. Laminarin is a mixture glucans [166] which is extracted from the common sea oarweed *Laminaria digitata*. Other effective crop immunity elicitors may be discovered from further purification of cell wall/secreted fungal PAMPs. 

#### 6.3.2. Discovery and Cloning of Resistance Genes in Crop Genomes

In the gene-for-gene hypothesis of plant immunity, a plant resistance (R) gene product is necessary for recognition of a pathogen avirulence (Avr) gene product. In many instances, the Avr gene encodes a secreted effector protein, which enables virulence on susceptible cultivars, but activates protective immunity and/or a hypersensitive response (HR) when an R gene is encoded in the host genome [25]. The HR is an effective strategy to inhibit disease which involves the rapid death of host cells at the site of infection, which may restrict fungal growth and dissemination to uninfected parts of the plant. 

Many R genes encode leucine rich repeat (LRR) regions, which in the ‘receptor-ligand’ model enable direct R protein interactions with specific pathogen effectors [167]. Alternatively, in the so called ‘guard model’, R proteins activate host immunity by detecting alterations in host target caused by the Avr protein [168]. Numerous plant-infecting fungal genomes are precited to encode hundreds of secreted protein effectors, and many of these may function as Avr genes in one or more resistant cultivars or species. Mapping and characterization of the cognate R genes in plant genomes (e.g., [169]) offers outstanding opportunities for biotechnologists to engineer resistant cultivars by cloning R genes into a desired cultivar/crop. As one example, expression of the pigeonpea R gene *CcRpp1* was able to confer full resistance to the devastating basidiomycete pathogen *Phakopsora pachyrhizi* in soybean [170]. These data demonstrate how non-commercially important plants could be an important resource for discovery and cloning of effective R genes in economically important crops. It should be noted that it is also possible to clone multiple R genes in so called stacking experiments, which is likely to provide increased durability of resistance when compared to single-gene transgenic crops [171,172]. In future, comprehensive understanding of secreted fungal Avr proteins, and their detection by host R proteins, may reduce crop losses to fungal disease, increase global food security, and minimize global fungicide applications.

#### 6.3.3. Inhibiting Secretion for New Mode of Action Antifungals

For human diseases caused by fungi, only three classes of drugs are clinically approved, including the echinocandins, azoles, and polyenes. These drugs perturb β-1,3-glucan synthesis (by inhibiting the glucan synthase Fks1), ergosterol biosynthesis (by inhibiting the lanosterol demethylase Erg11/Cyp51), and plasma membrane integrity (by directly binding ergosterol), respectively [12]. All clinical antifungals have significant side effects, and have reported reduced efficacy due to antifungal resistance [13]. Consequently, novel antifungal compounds are urgently required, especially those with new mode of actions.

Given that the classical secretion route is essential for polar growth (Section 2), nutrient acquisition (Section 3), and production of certain virulence factors (Section 6.1 and Section 6.2), targeting molecules to disrupt vesicle trafficking is a promising strategy to discover broad acting antifungals for use in the clinic. For example, screening of a >49,000 compound library revealed two hydrazine molecules which inhibited biosynthesis of the sphingolipid glucosylceramide in *C. neoformans* but not human cell lines [173]. These molecules were found to be active against a range of human infecting fungi, including *H. capsulatum*, *C. albicans*, and *A. fumigatus*. Microscopic analysis revealed that the antifungal activity of these molecules against *C. albicans* and *C. neoformans* mechanistically occurs via the disruption of Golgi and vesicle architecture [173,174]. Further analysis to identify putative drug targets used a combination of deep-sequencing and haploinsufficiency profiling/homozygous deletion assays in *S. cerevisiae* [175]. This revealed probable inhibition of various proteins associated with vesicle sorting or transport between Golgi apparatus and endoplasmic reticulum, including a subunit of the clathrin associated protein complex Apl5 [173,174].

Another study identified a new mode of action antifungal targeting secretion by growing bacteria isolated from marine invertebrates [176]. From over 1400 bacterial strains, a highly diverse chemical producing isolate was identified, which was prioritized for fermentation analysis. Using chromatographically purified compounds arrayed in over 170 96-well plates, a polyketide with high efficacy against multiple human infecting fungi was discovered, which was named turbinmicin [176]. Next, through chemical genetic screens using both *S. cerevisiae* and *C. albicans* mutants, turbinmicin was demonstrated to inhibit Sec14, a phosphatidylinositol/phosphatidylcholine transfer protein required for trans-Golgi functions [177]. The inhibition of secretion following turbinmicin exposure was further demonstrated using a GFP labelled v-SNARE vesicle membrane receptor protein Snc1. In budding yeast, this protein is trafficked through the classical secretory system and normally localizes to the plasma membrane at early bud sites in exponentially growing cells, but was observed to remain at the Golgi following turbinmicin treatment [176]. Encouragingly, this polyketide has low toxicity to human red blood cells, and reduced fungal burden in kidney and lung during murine models of *Candida auris* and *A. fumigatus*, respectively [176]. Taken together, recent targeting of Apl5 and Sec14 using newly discovered compounds demonstrate that molecules which inhibit protein product(s) in the classical secretory system are promising strategies to inhibit fungal infection of humans. 

Additionally, many secreted proteins which are located at the cell surface are attached to the plasma membrane by glycosylphosphatidylinositol (GPI) anchors. This induces numerous cell wall synthesizing and remodeling proteins. Chemical inhibition of the GPI biosynthetic pathway is therefore an promising approach, firstly as it leads to lysis via loss of cell wall integrity [178], and secondly as defective cell wall synthesis exposes glucans to recognition by the host [179]. Also encouraging is the limited conservation between fungal and human enzymes in the GPI biosynthetic pathway. One recent example of antifungal discovery via GPI inhibition harnessed a luciferase gene reporter fused to a *C. albicans* GPI-anchor signal in *S. cerevisiae* to screen for chemical inhibitors of the GPI pathway [180]. This revealed that a natural bacterial product, termed jawsamycin, efficiently inhibited GPI synthesis by targeting a subunit of the fungal UDP-glycosyltransferase Gpi3, which mediates the first step in GPI biosynthesis [180]. Encouragingly, this molecule did not inhibit the human orthologue, PIG-A but reduced fungal burden and elevated survival in murine models of *Rhizopus* infections [180], suggesting jawsamycin is an exciting lead compound for further applications of GPI inhibition.

#### 6.3.4. Secreted Molecules and Vaccines for Fungal Infection

Although there is currently no licensed vaccine for prevention of fungal disease in humans, some secreted molecules have been incorporated into candidate vaccines and tested for protection in murine models and early clinical trials. For example, the Candida spp. agglutinin-like sequence proteins are a family of secreted cell surface molecules which mediate adhesion, including to abiotic surfaces, human endothelial/epithelial cells, and other microbes (reviewed in [181]). Als proteins contain secretion signals and GPI anchors, and facilitate invasion of host cells during disease [182]. Excitingly, the N-terminus of Als3 with aluminum hydroxide adjuvant has been demonstrated to safely induce protective antibody and cell-mediated immune responses in humans in a phase one clinical trial [19]. An updated vaccine (NDV-3A) developed by NovaDigm™ Therapeutics is currently under phase 2 clinical development against recurrent vulvovaginal candidiasis. Recent work has also demonstrated that NDV-3A is protective against the emergent pathogen *C. auris* in murine models of infection [20]. 

Elsewhere, prevention of candidiasis in mouse models of kidney infection has been achieved by immunization with secreted aspartic protease 2 (Sap-2) [21]. *Candida* saps cause disruption to host tissue [183] and degradation of the host complement system [146]. More recent incorporation of truncated Sap-2 into a virosomal vaccine, called PEV7, against candidal vaginitis was also protective in rat models [184], and was effective at generating B-cell-mediated immune responses in phase 1 clinical trials [185]. Various other extracellular molecules have been applied for vaccines against *Candida* spp., *Pneumocystis* spp., *A. fumigatus*, and *C. neoformans*, including a range of cell wall polysaccharides (recently reviewed in [186]). Taken together, the utilization of several secreted fungal molecules during human infection may enable the development of an effective vaccine in the near future.

### 6.4. Summary

Secreted fungal molecules can be crucial mediators of fungal diseases, and thus provide many avenues for therapeutics in the clinic or fungicides in agriculture. Increased exploration of the infectious arsenal secreted by invading fungi will offer future avenues for developing vaccines, new mode of action chemistries, and disease resistant crop varieties. Finally, secreted pathogen molecules can be used for timely diagnosis of disease, thus enabling the most effective therapeutic interventions when pathogen damage and proliferation are low.

## 7. Interspecies Communication

Despite their critical impact in agriculture and the clinic, disease causing filamentous fungi are the exception, with many more species existing either as saprobes or mutualists. As is the case during infection, establishing and maintaining symbiotic and mutualistic interactions are largely facilitated by filamentous growth and secretion, although in many instances, the precise fungal molecules, and their translocation and functional role upon or inside other organisms, remain unclear. The most notable examples are arguably lichens, a composite organism of algae or cyanobacteria (photobiont) which live among a filamentous fungus (mycobiont), and, additionally, arbuscular mycorrhiza, a root symbiosis between Glomeromycetes fungi and approximately two thirds of terrestrial plants. The role of secreted fungal molecules in these symbiotic and mutualist life modes, and their potential applications, are summarized below.

### 7.1. Fungal Extracellular Molecules in Mutualist Interactions: Lichens

Lichens are extremely common mutualist organisms which are estimated to cover 8% of terrestrial land surfaces [187], with approximately 20% of all fungi being lichen formers [188]. Filamentous growth (and therefore secretion, see Section 2) is absolutely critical to the formation of the lichen body, termed the thallus. In this structure, for example, tightly packed hyphae form a cortex at the exposed surface, under which loosely packed hyphae can be inhabited by the photobiont [187]. In many instances, filamentous ascomycete fungi require basidiomycete yeasts for formation of the cortex [189]. Lichen phytobionts gain carbohydrates from photosynthesis, and mycobionts acquire water, nutrients, and a solid substratum for growth that protects from desiccation [190].

Transcriptional analysis suggests that secretion of fungal effectors and secondary metabolites are crucial for the initiation of the mutualist lifestyle. For example, gene expression profiling of mycobiont *Cladonia grayi* and algal photobiont *Asterochloris glomerata* co-cultivation revealed transcriptional deployment of fungal small molecular weight, putatively secreted proteins (identified by classical N-terminal secretion signals), extracellular hydrolases, and membrane transporters [191]. When combined with expansion of predicted polyketide synthase encoding genes in the *C. grayi* genome [191], these data suggest that, as is the case with pathogenic fungi, the secretion of effector proteins and secondary metabolites via the classical route and/or membrane transporters, respectively, may mediate lichenization. Recent metagenomic analysis of lichens also demonstrates expanded secondary metabolite production capacity (especially in the ascomycete member), and, additionally, a gene repertoire that is predicted to produce acidic extracellular polysaccharides (e.g., glucuronoxylomannan) which may be required for the formation of an extracellular matrix that physically links cells for mutualistic interactions in the thallus [192].

Due to the wide variety of secondary metabolites identified from lichens, these organisms can be viewed as promising natural product libraries for new bioactive molecule discoveries. Indeed, over 300 secondary metabolites have been characterized from lichens, including depsides, depsidones, dibenzofurans, xanthones, among others, with antibacterial, antiviral, antifungal, insecticidal, antiangiogenic, anticancer, and other properties (reviewed in [190,193,194]). An obvious challenge to the use of lichens as microbial cell factories to produce these metabolites is slow growth, technical difficulty of in vitro culture, and a lack of molecular tools [193]. However, an attractive solution which may obviate all these limitations is heterologous expression of secondary metabolite biosynthetic genes in chassis fungi (e.g., *A. niger* see Section 9.1). As the number of sequenced lichen-forming fungal genomes increases, and tools necessary for their heterologous expression become more widespread (e.g., polycistronic fungal gene expression cassettes for multiple secondary metabolite biosynthetic genes [195]), it is exciting to speculate that the chemical basis of fungal mutualists may be exploited for a variety of bioprospecting applications in the future.

### 7.2. Arbuscular Mycorrhizae and Fungal Secretion

Fungi from the phylum Glomeromycota form obligate symbiosis with plant roots that have fundamental roles in terrestrial ecosystem function and agriculture. Mycorrhization results in more efficient inorganic and micronutrient uptake from the soil for the plant, while fungi obtain sugars and lipids from the host [196]. For establishment of arbuscular mycorrhizae, hyphal filaments grow along chemical gradients in the soil and physically penetrate the root epidermis using appressorium, a pressure-based infection structure best characterized in the rice pathogen *M. oryzae* [197]. After entry to the host parenchyma, hyphae grow to form specialized nutrient exchange structures termed arbuscules [198]. Other mycorrhizal associations exist, including ectomycorrhiza, where fungal hyphal do not penetrate the root but rather grow upon its surface.

As is the case with plant-infecting fungi, arbuscular mycorrhizal genomes are enriched with hundreds of small molecular weight, cysteine rich, putative effector proteins which encode N-terminal secretion signals [199]. Comparative genomics across the Glomeromycota confirm that while many effectors lack robust functional predictions, they are highly conserved in this phylum [199], suggesting they play a crucial and fundamental role in symbiosis. Functional analysis of symbiont effectors has demonstrated that they include secreted LysM effectors, which bind fungal chitin and thus reduce plant immune responses [200]. Others co-localise to with the plant nucleus, suggesting that they modify host gene expression during arbuscule development [201]. Thus, it is likely that a key commonality between plant pathogenic and symbiotic fungi is the secretion of protein effectors for modulating plant immunity to enable filamentous growth [202].

The most obvious applications for arbuscular mycorrhizae is in agriculture, with fungi acting as biostimulants (e.g., increasing crop stress tolerance), biofertilizers (e.g., increasing nutrient availability) and biopesticides/biocontrol agents (e.g., reduce crop loss from infectious disease, reviewed in [203]). It seems clear that full understanding of the plant mycobiome, and how this is mechanistically mediated by secreted effector proteins, will enable the development of renewable, environmentally inert agricultural technologies. 

### 7.3. Summary

The function of many fungal secreted molecules during symbiotic and mutualistic interactions is currently unclear, yet this knowledge has crucial implications for our understanding of natural ecosystems and agriculture, which ultimately may translate to many applied benefits. In this section, we have highlighted lichens and arbuscular mycorrhizae as natural product libraries and break through crop technologies, respectively. However, the many other symbiotic or mutualistic interactions between fungi and other organisms, e.g., in marine ecosystems [204] or in human disease [205], are beyond the scope of this review. Nevertheless, it is clear that secretion is fundamental for the establishment and maintenance of symbiotic and mutualistic fungal interactions.

## 8. Intraspecies Communication

Many secreted fungal molecules are thought to play intraspecies signaling roles to orchestrate growth and development in the natural niche. These may be between genetically distinct members of the same species or clonal components of the same isolate. Additionally, secretion may be used to outcompete of defend against other species (see Section 9). Collectively, these extracellular molecules have been described as the chemical language of fungi [206], and include the production of secondary metabolites/volatile organic compounds such as germination autoinhibitors, chemoattractants which mediate hyphal fusion during mycelial formation (anastomosis), sex pheromones, and other signaling molecules which play many unknown and specialist roles. Such decoding of the fungal language ultimately enables it to repurposed in applications from drug discovery to disease diagnostics. In this section we summarize current understanding and potential applications of two types of secreted fungal molecules sex pheromones and antifungal proteins which mediate intraspecies communication, growth, and development.

### 8.1. Sexual Reproduction and Secretion of Pheromones

Fungal mating can occur by heterothallic sex, where a zygote is formed from nuclei contributed by two different individuals (outcrossing). Alternatively, in homothallic sex, both nuclei are derived from the same strain (selfing). Many filamentous fungi are predicted to sexually reproduce based on the presence of a sex-specific region in the genome termed the mating-type locus (MAT), which encodes transcription factors necessary for regulating expression of mating and cell-type genes [101,207]. Distinct, yet compatible mating types are needed for heterothallic sexual reproduction, whereas homothallic sex occurs via universally compatible types [208]. The frequency of sexual reproduction may be very rare compared to asexual growth [209], which is thought to result from a balance of the costs and benefits (reviewed in [208]). One of the main benefits of sex is recombination during meiosis, which can aid adaptation by combining beneficial mutations into a single genome, or, alternatively, separating beneficial from deleterious ones. A significant cost is the necessity to find a compatible mate and the energy/growth costs of producing reproductive structures.

Secretion plays a vital role in fungal sex, as extracellular pheromones mediate chemotrophic growth between pairs of heterothallic or homothallic cells. In mucoraceous fungi, for example, the sex pheromone is the secondary metabolite trisporic acid [30]. Although the mode of export is not precisely characterized, it presumably occurs via membrane transporters [30]. In many ascomycete fungi, protein pheromones are exported via the classical route (Section 2) as evidenced by the presence of peptide secretion signals in several protein sequences, for example PpgA in *Aspergillus* spp. or MFALPHA in *Candida* spp. [101]. Much work from model fungal systems, including *S. cerevisiae, N. crassa*, and *A. nidulans* has revealed the mechanisms behind signal transduction of protein pheromones, which activate G protein-coupled receptors (GPCR) at the surface of the cell [210]. The pheromone response is transduced via signaling cascades, notably the mitogen-activated protein kinase (MAPK) pathway. In *S. cerevisiae*, for example, the response to pheromone includes activation GPCR Ste2, MAPK signaling and induced expression of mating and filamentous growth genes via the transcription factor Ste12 [211], polar growth along the pheromone gradient, and G1 cell cycle arrest which enables nuclear fusion (reviewed in [210]). In *Aspergillus* spp., including *A. nidulans*, *A. fumigatus*, and *A. flavus*, the MAPK pheromone response pathway includes three kinases (SteC, MkkB, and MpkB), an adaptor (SteD), and the scaffold (HamE) [212,213].

### 8.2. Applications of Fungal Pheromones

In many filamentous fungi, sexual reproduction is linked with secondary metabolism. For example, in *Aspergillus* spp., the conserved nuclear velvet protein complex consists of VelB, VeA, and LaeA which controls development (e.g., by repression of asexual growth through inhibition of the transcription factor VeA) and activation of secondary metabolite biosynthetic gene clusters (e.g., chromatin remodeling by the methyltransferase LaeA) [214]. Activation of the pheromone MAPK pathway in *Aspergillus* spp. causes the MAPK Fus3 to translocate into the nucleus and activate both sexual and secondary metabolite processes via transcription factors such as VeA (recently reviewed in [215]). Consequently, stimulating sexual development and secondary metabolism via synthetic stimulation of the pheromone response pathway components may be an outstanding avenue for elevating titres of natural products or during drug discovery. For example, the heterothallic mould *Blakeslea trispora* is used to ferment the vitamin A precursor and food dye β-carotene [216]. Titres of this molecule can be increased during industrial fermentation either by inoculating complementary mating types in bioreactors, or through the addition of the pheromone trisporic acid [30].

Recent work has also demonstrated that secretion of pheromones can be repurposed in a biosensor-based diagnostic for fungal disease [9]. In this work, the yeast GPCR encoding gene *ste2* was replaced with orthologues from human and plant infecting fungi (e.g., *C. albicans*, *P. brasiliensis*, *H. capsulatum, Botrytis cinerea*, *Fusarium graminearum*, *M. oryzae*). Additionally, a simple read out of GPCR activation was developed, whereby genes necessary for biosynthesis of the red plant pigment lycopene were constitutively expressed and a single gene placed under control of the pheromone inducible gene *fus1*. Thus, the production of a red colony, which was visible by eye in a dipstick assay, was indicative of GPCR activation and pathogenic growth for a range of filamentous and dimorphic fungi [9]. These authors therefore demonstrate that GPCR activation by pheromones can be used to detect growth of infectious fungi in both agriculture and the clinic.

### 8.3. Applications of Antifungal Proteins

Numerous ascomycete fungi secrete small molecular weight, negatively charged, cysteine rich proteins, which have conventionally been identified by their antifungal activity in various growth assays. Named accordingly, antifungal proteins (AFPs) are secreted during low growth phases by *Aspergillus giganteus* (AFP), *A. niger* (AnAFP), *P. chrysogenum* (PAF), *Penicillium digitatum* (AFPB), and *Neosartorya fischeri* (NAFP), with over 50 orthologues present in various ascomycete genomes [22,23].

N-terminal signal peptides strongly suggest that AFPs are secreted via the classical route (see Section 2), many as inactive pro-peptides which require cleavage by extracellular proteases for antifungal activity [23]. The mode of antifungal action has been unveiled for several AFPs and includes inhibition of chitin biosynthesis, calcium signaling (reviewed in [22]) and activation of the cAMP/Pka signaling cascade via the heterotrimeric G-protein signal transduction pathway [217]. Conventionally, these proteins were assumed to act as antifungal molecules to kill competing fungi in the natural niche. However, several lines of evidence suggest that these secreted proteins could actually function as cannibal toxins important for asexual colony development in their producing hosts strains by killing genetically identical cells [22,218]. Firstly, AFP expression is precisely controlled both temporally and spatially, occurring predominantly in vegetative mycelium of filamentous fungi (reviewed in [22]). Deletion or overexpression of AFPs results in a drastic reduction of conidia in *P. chrysogenum* and defective vegetative growth in *P. digitatum,* respectively [219,220], which is consistent with a role in asexual development of the host. Additionally, in *A. niger*, transcriptional meta-analyses revealed that AnAFP is co-expressed with multiple genes associated with asexual development and response to carbon limitation [221,222], the latter being a key nutritional cue which induces conidiation in ascomycetes. Additionally, AnAFP is regulated by various transcription factors which are critical for vegetative growth, including the developmental regulator StuA and VelC [221]. More recent work in *P. digitatum* demonstrated that AFPB interacts with the producing host strains cell wall, is internalized by energy dependent mechanisms, and subsequently activates gene expression necessary for programmed cell death [218].

Given their effective inhibition of fungal growth and viability, AFPs have are promising candidates for developing novel therapeutics. Due to their specific mode of action, they have high efficacy against susceptible fungal species and low toxicity to both plant and mammalian cells and tissues [22,223]. As one example of their potential applications, transgenic rice expressing AFP demonstrated enhanced resistance to the *M. oryzae* infection [224]. Additionally, as several AFPs seem to function by multiple mechanisms (e.g., concomitantly perturbing cell wall activity and activating cell death), it is reasonable to speculate that antifungal resistance will be more difficult to develop when compared to single mode-of-action chemistries.

### 8.4. Summary

Intraspecies communication in filamentous fungi fundamentally controls filamentous growth, for example by pheromone-based chemotaxis or AFP programmed cell death. While we have highlighted some applications of these secreted molecules, many other self-communicators also have putative technological benefits, for example, inhibition of pathogenic spore germination by autoinhibitors or their synthetic analogues. We predict that improved understanding of this component of filamentous fungal language will enable many technological breakthroughs in the future.

## 9. Surviving in the Community

In the natural niche, filamentous fungi interact with many organisms, including resource competition with other fungi or bacteria, or predation from amoeba and insects. Many secreted proteins, acids, and secondary metabolites may function to attack, defend against, and manipulate other community members, thus producing a wealth of bioactive molecules with diverse applications (recently reviewed in [225]). Consequently, moving from conventional axenic to mixed laboratory cultures is a highly promising strategy to activate the biosynthesis of previously undescribed fungal molecules, most notably secondary metabolites [226]. In this section, we summarize exemplar fundamental insights and applications for secretion which may play direct role(s) in filamentous fungal survival in mixed communities, including repurposing fungal antimicrobial production for new bioactive molecule discovery, harnessing and inhibiting membrane efflux pumps, and considering if organic acid secretion could be a strategy to kill competing microbes.

### 9.1. Harnessing Fungal Chemical Defence for New Bioactive Molecule Discovery: A Case Study Using Enniatin and Related Compounds

The most successful repurposing of fungal offensive chemical weaponry is the use of beta-lactam antibiotics in medicine. These molecules are likely used in the natural niche to inhibit growth of competing bacteria by disruption of cell wall synthesis. However, the fungal chemical repertoire is vast, and numerous new classes of bioactive molecules have recently been discovered and manipulated for new applications. For example, enniatins are a group of secreted cyclohexadepsipeptide secondary metabolites isolated from *Fusarium* moulds with a range of bioactivities, including antiinsectan, antifungal, antibacterial and anticancer properties [31]. Experimental evidence suggests that one of their natural functions may be to prevent consumption of the fungus by insects, for example by beetles when growing as a contaminant on wheat grains [227]. Enniatins have been developed into the antimicrobial and anti-inflammatory throat spray fusafungine [228], supporting the hypothesis that these molecules and their derivatives are promising for therapeutic applications. In order to generate a highly efficient expression host, the enniatin NRPS gene *esyn1* from *F. oxysporum* was codon optimized and expressed in the cell factory *A. niger* [229]. Through optimization of culture conditions, and additional expression of a second enniatin biosynthetic gene (which was necessary for the synthesis of the precursor d-2-hydroxyvaleric acid, d-Hiv), highly efficient production of enniatins was possible in the heterologous host (up to 4.5 g/L, [229]). Subsequent studies have used *A. niger* for heterologous expression of other cyclic peptide secondary metabolites, including the beauvericins and bassianolide [230]. Interestingly, by adding different analogues of the d-Hiv precursor to growth media, beauvericin derivatives could be generated, such as bromo-beauvericin [230]. A more recent study which also generated new-to-nature cyclodepsipeptide derivatives in *A. niger* conducted domain swap experiments of NRPS genes to produce compounds octa-enniatin and octa-beauvericin (at mg/L concentrations) and hexa-bassianolide (at g/L concentrations) [231]. Excitingly, two hybrid molecule peptides demonstrated higher antiparasitic activity when compared to their naturally occurring compounds. Moreover, both were several times more efficacious when compared to existing therapeutics, e.g., benznidazole activity against *Trypanosoma cruzi* [231].

As reviewed recently [225], the chemical strategies which filamentous fungi use to protect and survive in mixed communities are hugely diverse. Indeed, many genomes contain biosynthetic gene clusters for several dozen putative metabolites (e.g., over 80 in *A. niger* [232]), and most of these are transcriptionally silent during in vitro culture [233]. Therefore, the above studies which developed highly efficient heterologous expression hosts and methods for generating new-to-nature derivative molecules pave the way for future repurposing of secreted fungal metabolites in drug discovery.

### 9.2. Secretion of Xenobiotics: Surviving Chemical Attack

In the natural niche, filamentous fungi are constantly exposed to toxic compounds, which can be synthesized by competing microbes or by the host during disease. One of the most fundamental functions of secretion is therefore to export toxic molecules from the cell, thus preventing intracellular accumulation and enabling resistance. Fungi have evolved a remarkable diversity of plasma membrane drug efflux pumps, including ATP-binding cassette transporters (ABC transporters) which use ATP to translocate the substrate, and major facilitator superfamily (MFS) transporters which move small solutes via chemiosmotic gradients. Several dozen or even hundreds of ABC and MFS transporter genes are typically encoded in a single fungal genome, many of which may act as drug efflux pumps [234,235,236]. Additionally, these proteins may be necessary for the export of specific molecules or may function as pleiotropic transporters with broad substrate specificity, making them highly effective at cell detoxification. These xenobiotic membrane transporters are of major interest, both for generating hypersecretion isolates in biotechnology, and for combating antifungal resistance in medical and agricultural settings.

### 9.3. Applications of Fungal Efflux Pumps

In addition to protecting against xenobiotic compounds, efflux pumps play a crucial role in protecting the producing host fungus from self-toxicity. Given that many secondary metabolites are promising for new bioactive molecule discovery (see above), the controlled expression of efflux pumps is a promising strategy to maximize product titres, which is highly comparable to approaches used for organic acids [132,237].

In some instances, identification of the cognate transporter for a molecule of interest can be achieved by delineation of secondary metabolite BGCs on the fungal genome. BGCs commonly consist of a gene encoding a core biosynthetic enzyme (e.g., a NRPS or polyketide synthase), which are contiguously clustered with genes encoding (i) tailoring enzymes, which modify the nascent molecule; (ii) transcription factors for regulation of BGC expression, (iii) and secondary metabolite plasma membrane transporters for molecule efflux from the cell. One of the best characterized BGC is necessary for gliotoxin production by *A. fumigatus*, which includes an NRPS (*gliP*), transcription factor (*gliZ*), numerous tailoring genes, and the membrane transporter *gliA* [238]. Disruption of *gliA* in *A. fumigatus* increases susceptibility to exogenous gliotoxin, and reduces export of this molecule from the cell [238]. Additionally, *gliA* overexpression in *S. cerevisiae* also elevated resistance to exogenous gliotoxin [238]. Therefore, screening fungal genomes for putative drug transporters within BGCs may be an effective strategy to identify and functionally characterize secondary metabolite transporters in order to generate hyperproducing strains.

It should be noted, however, that in some instances transporter genes founds within a BGC are functionally redundant. For example, deletion of the putative MFS transporter gene *aflT*, which is located in the aflatoxin BGC, did not impact export of this metabolite in *Aspergillus parasiticus* [239]. One exciting possibility is that pleiotropic MFS or ABC transporters, which export multiple secondary metabolites from the cell, could be used to generate hypersecretion of multiple products in a single mutant background. These transporters must exist in fungal genomes, as evidenced by heterologous expression systems efficiently secreting xenobiotic metabolites in pilot fermentation studies (see enniatins above).

Antifungal resistance is a persistent problem both in agriculture and the clinic [13]. It is possible that the use of certain fungicides for crop disease, notably azoles, may cause elevated resistant to this class of antifungals in medical settings [240]. Resistance can occur by multiple mechanisms, including mutations in the target enzyme and via the overexpression of multidrug efflux pumps, such as the MFS transporter Mdr1 in *C. albicans* or various ABC transporters in *A. fumigatus* [241,242]. Encouragingly, structural aspects of these membrane transporters may be unique to fungi, such as the presence of two extracellular loops between transmembrane spanning regions in fungal ABC transporters [242]. Thus, the chemical inhibition of efflux pumps should theoretically have minimal toxicity to humans and may act to improve efficacy of existing antifungals when used in combinational therapy. Several studies have screened compound libraries for inhibitors of fungal multidrug efflux pumps. Heterologous expression of efflux pump encoding genes is an attractive option compared to assays using clinical isolates, as the latter strains may possess multiple mechanism of antifungal resistance (reviewed in [243]). In *S. cerevisiae*, the deletion of many predicted efflux pump encoding genes in a single strain has been conducted to generate an improved heterologous expression host for such assays [244], and the inhibition of drug transport can be easily screened in high throughput. Screens have included natural product libraries [245], small molecule libraries [246], or hypothesis-driven libraries based on predicted transport inhibitors, e.g., octapeptides [247]. Many of these screens have been successful in generating new leads for chemical inhibition of fungal efflux pumps, including enniatins (see above), octapeptides, and cerulenin derivatives. In summary, increased understanding of the genetic and mechanistic basis of fungal drug efflux and the chemical inhibitors of these proteins may enable effective control of antifungal resistance in future.

### 9.4. Revisiting Organic Acids: Are They Secreted to Inhibit Competing Neutrophiles?

The above sections have discussed evidence that certain organic acids may be secreted to chelate micronutrients (Section 4) or to rebalance intracellular metabolism by removal of overflow metabolites (Section 5). An additional and mutually not exclusive hypothesis is that acidification of the external environment may inhibit the growth of other organisms which cannot intrinsically tolerate low pH. While many fungi are capable of producing organic acids and growing at low pH [248], direct evidence supporting the competing microbe hypothesis is limited, with minimal experiments regarding fungal organic acid secretion conducted in mixed cultures [226]. It is interesting, however, that certain other systems use organic acids as antibacterials. For example, mammalian macrophages produce itaconic acid to perturb pathogen metabolism [249], and the degradation of this acid is required for full virulence of some bacteria [250]. Thus, it is conceivable that, in some instances, TCA derived organic acids may be secreted as a generalist antimicrobial weapon in the natural niche. It is interesting to speculate that the production of certain organic acids during fermentation may become more efficient in future by either co-cultivation [226] of synthetic activation of antimicrobial fungal defences.

### 9.5. Summary

Free living, pathogenic, and symbiotic fungi must survive amongst a range of competing and/or predatory organisms. We have summarized only three aspects of this secrete-to-survive strategy with bioactive secondary metabolites, detoxification via efflux pumps, and killing via organic acids as exemplars for a staggeringly diverse armamentarium of defensive and offensive fungal capabilities. Clearly, applications to reengineer and apply such secretion will provide major benefits to medicine, agriculture, and industrial biotechnology. Consequently, a detailed mechanistic understanding of how and why fungi secrete to survive in the natural niche will enable significant applied breakthroughs in the future.

## 10. Concluding Remarks

Filamentous fungal secretion is as varied as the many lifestyles and niches in which these organisms thrive. Indeed, given the role of secretion in polar growth and nutrient acquisition, secretion can be viewed both as something that fungi do, and also essentially what they are: a chassis for selectively turning the inside out, and then the outside in. This review has covered secretion from an interdisciplinary and non-anthropocentric viewpoint, and as such cannot summarize the staggering diversity of fungal secretion and its many applications in bioeconomy, human health, food security, and many other aspects of life on earth. Nevertheless, we argue that understanding secretion across disciplinary boundaries both in the applications and the class of extracellular molecules will facilitate rapid methodological and technological breakthroughs which will enable the full potential of filamentous fungi to be realized.

## Figures and Tables

**Figure 1 jof-07-00535-f001:**
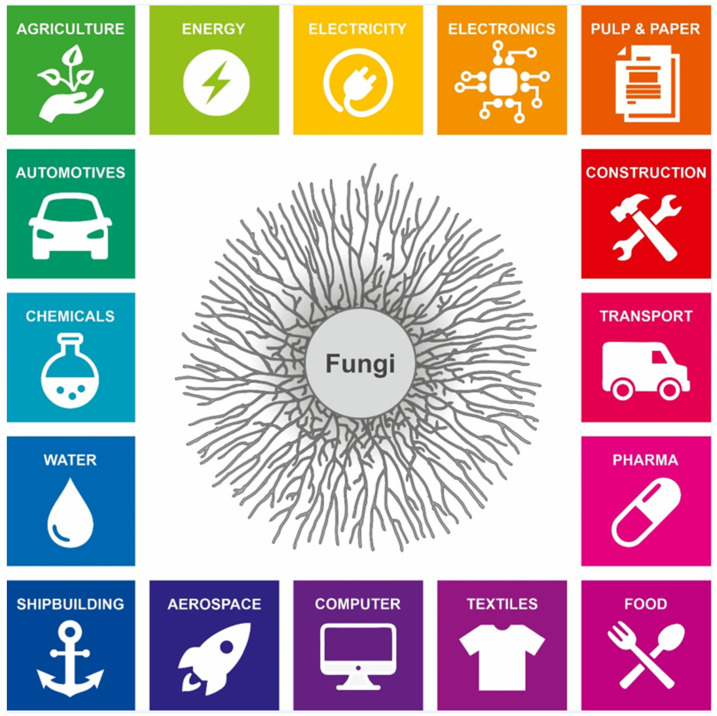
Industries utilizing the metabolic capacities of filamentous fungi. Reproduced from [6], CC BY 4.0.

**Figure 2 jof-07-00535-f002:**
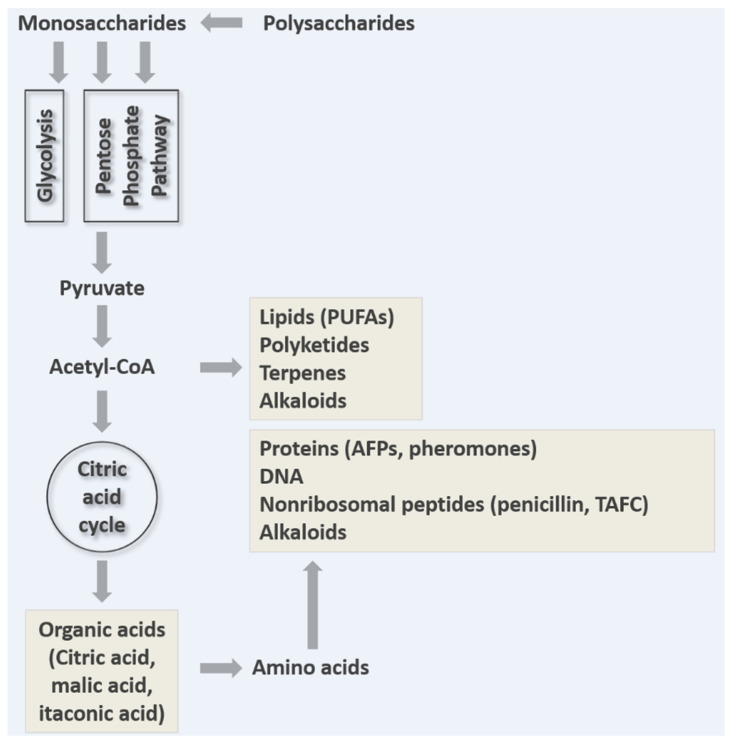
Interconnections between organic acid, protein, and secondary metabolite biosynthesis in filamentous fungi. Schematic diagram is a simplified model of filamentous fungal carbon catabolism when utilizing sugars. The main classes of molecule are depicted in grey boxes, with some exemplar secreted molecules discussed in this review highlighted in parentheses. Acetyl-CoA is a crucial link between both primary and secondary metabolism. Note ATP and NAD(P)H are not indicated in the schematic for simplicity. Modified from [14].

**Figure 3 jof-07-00535-f003:**
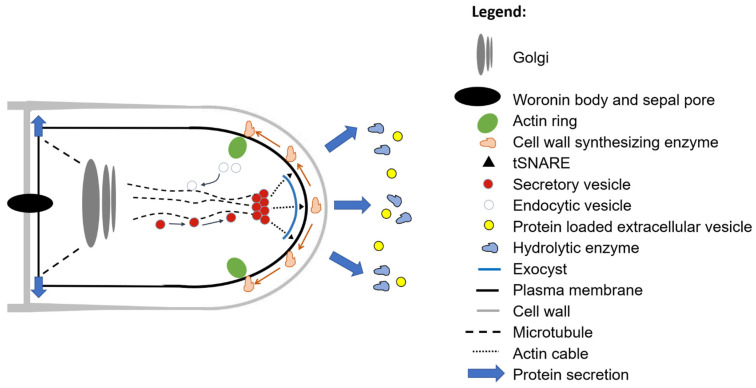
Schematic representation the filamentous fungal protein secretory system. In the classical route, cargo loaded vesicles are transported from the Golgi along microtubules and actin cables to the tip, where they are fused to the plasma membrane by the exocyst complex. Growth of the tip moves cell wall synthetic enzymes toward intercalary hyphal regions (orange arrows), where they are putatively recycled at the actin ring and the endosome system to maintain polarity. Protein secretion does occur at intercalary regions, possibly including the septa. Not shown is secretion by non-classical route(s) which may occur by as yet uncharacterized plasma membrane transporters. The production of extracellular vesicles is also not yet fully understood. Note that key differences to the secretory machinery (e.g., protein components of SPK, exocyst) exist between phyla, and are still being elucidated. Note that not all components are show and diagram is not to scale. Adapted from [3].

**Figure 4 jof-07-00535-f004:**
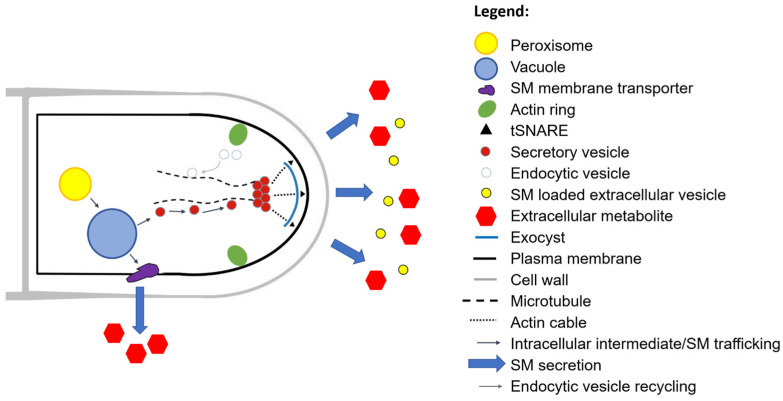
Schematic representation of secondary metabolite secretion by filamentous fungi. Generation of metabolite intermediates occurs in various subcellular locations, including peroxisomes and vacuoles. Secretion of certain secondary metabolites (SM) may be coupled with the classical secretory route. Alternatively, plasma membrane transporters may be required for export from the cell. Where these transporters are localized on the hyphae (e.g., tip, intercalary region, septa) is unknown for the majority of transporters. SM associated biosynthetic enzymes have been detected in extracellular vesicles in some instances. SM may themselves be packaged in extracellular vesicles. Note that not all components are show and diagram is not to scale. Adapted from [3].

**Figure 5 jof-07-00535-f005:**
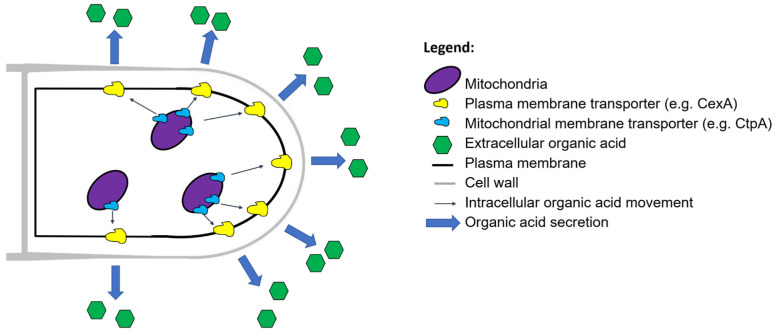
Schematic representation organic acid secretion by filamentous fungi. Many organic acids are intermediates of the Krebs cycle in the mitochondria. Under certain conditions, membrane transporters export organic acids firstly into the cytosol and then across the plasma membrane. Several such transporters have now been functionally characterized, but in general their location on the hypha (e.g., tip or intercalary regions) is unknown. Note that not all components are show and diagram is not to scale. Adapted from [3].

**Table 1 jof-07-00535-t001:** Summary information of filamentous fungal secretion and exemplar applications of their extracellular molecules. Species listed are meant as examples only and are not exhaustive. The most common postulated and/or confirmed functions of these molecules are given, but in several instances their precise role is still unclear (e.g., citric acid).

Product Class	Product	Exemplar Producing Fungal Species	Secretion Route and/or Plasma Membrane Transporter	Postulated or Confirmed Function(s) in the Natural Niche	Current or Prospective Application(s)	References
Enzyme/proteins	Glucoamylase GlaA	*A. niger*	Classical secretion route	Nutrient acquisition	Food and beverage industries processing starch to glucose	[4,15]
	Cellulases	*T. reesei*, *T. thermophila*			Textile, food, and other industries	[16]
	Lipases	*T. lanuginosus* *A. oryzae*			Applications in biodiesel, dairy, textile, detergent, paper, pharmaceutical, leather and other industries	[17,18]
	Agglutinin-like sequence protein Als3	*Candida* spp.		Adhesion of fungal cells to host and abiotic surfaces	Possible vaccine in humans	[19,20]
	Secreted aspartic protease 2 Sap-2			Nutrient acquisition, virulence factor		[21]
	Antifungal protein AFP	*A. giganteus*		Cannibal toxin regulating clonal growth, inhibitor of competing fungi	Prospective antifungal use in clinic or agriculture	[22,23]
	Effectors	*M. oryzae*, *Z. tritici*, other plant-infecting fungi	Classical secretion route, biotrophic interphase complex, non-classical secretion route, extracellular vesicles	Virulence factors	Cognate receptor and R proteins used to elicit protective crop immunity	[24,25,26]
Secondary metabolite	*Triacetyl* *-* *fusarine C*	*A. fumigatus*	Intermediates generated in mitochondria and peroxisomes plasma membrane transporter unknown	Micronutrient acquisition (iron chelator), necessary for full virulence	Biosynthetic pathway is a possible drug target, siderophores could be used to decontaminate metals from natural ecosystems	[27,28]
	Penicillin	*P. chrysogenum*	Intermediates generated in peroxisome and vacuoles, possibly secreted via vesicles or unknown plasma membrane transporter	Inhibitor of competing microbes	Antimicrobial with widespread clinical use	[29]
	Trisporic acid	*B. trispora*	Unknown	Mating pheromone	Used to elevate yields of β-carotene during fermentation	[30]
	Enniatin	*Fusarium* spp., *A. niger* (heterologous expression host)	Unknown	Prevent predation	Antimicrobial and anti-inflammatory clinical applications	[31]
Organic acid	Citric acid	*A. niger*	Plasma membrane transporter CexA	Micronutrient acquisition (iron chelator), inhibitor of competing microbes, overflow metabolite	Food and beverage as flavour enhancer, cleaning product, many industrial uses as a weak acid and platform chemical	[32]
	Malic acid		Plasma membrane transporter DctA	Inhibitor of competing microbes, overflow metabolite	Industrial uses as a weak acid and platform chemical	[33]
	Itaconic acid	*A. terreus*, *U. maydis*, *A niger* (heterologous expression host)	Plasma membrane transporter MfsA (*A. terreus*) or Itp1 (*U. maydis*)	Inhibitor of competing microbes, overflow metabolite	Industrial uses as a weak acid and platform chemical	[34]

## Data Availability

Not applicable.

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
