# Peer review of "Turning Inside Out: Filamentous Fungal Secretion and Its Applications in Biotechnology, Agriculture, and the Clinic"

_jof, 2021, doi:10.3390/jof7070535_

Round 1

Reviewer 1 Report

The manuscript “Turning inside out: filamentous fungal secretion and its applications in biotechnology, agriculture and the clinic” by  Timothy C. Cairns, Xiaomei Zheng, Ping Zheng, Jibin Sun and Vera Meyer summarizes the ocean of knowledge on fungal secretion. The article reviews more than 200 papers, some of them co-authored by the authors. The manuscript structure is clear and examples of genes/protein products are well suited to describe the main mechanisms described. I find it particularly useful for inexperienced mycology adepts to get the basics and be directed to the relevant literature. 

My main concern is the graphics, both figures do not relate well with the manuscript whereas several topics would be much more informative when illustrated. Let me explain this suggestion. Instead of Fig. 1. one could show the outline of the review which is a conscious selection of topics by the authors. Fig.2. would benefit from a cellular context, the compounds could be placed within a hyphae, with dedicated transporters, Golgi structures, SPK, etc. 

Moreover, I strongly suggest showing differences between Dikarya and non-Dikarya fungi, Saccharomycotina and Pezizomycotina in chapter 2 on canonical secretion, endocytosis and extracellular secretory vesicles. 

I would find it useful to present a graphical overview of types of molecules obtained from fungi with their secretion mechanisms, transporters, and the actual organism used in the industry to produce this compound.

Authors may decide to focus on Dikarya alone leaving away the issue of SPK and cytoskeleton differences. Otherwise, it would be recommendable to give at least a hint that GPCR repertoire, actin, dynein and other filament types are different in flagellated fungal lineages (eg. Kiss, EnikÅ‘, et al. "Comparative genomics reveals the origin of fungal hyphae and multicellularity." Nature communications 10.1 (2019): 1-13.). 

If Mucorales are to stay it is essential to underline they have a different cell wall composition (Bartnicki-Garcia, Salomon. "Cell wall chemistry, morphogenesis, and taxonomy of fungi." Annual Reviews in Microbiology 22.1 (1968)) what implies PAMP repertoire (glucan!). Their structure has some resemblance to the SPK but it does not necessarily have all the identical proteins out there.

It would be great if authors could provide at least a suggestion of lipid and alkaloids secretion examples and applications/significance. I found only carotenoids and trichosporic acid as the terpenoid examples and no lipids, for instance PUFA. One or two sentences summarizing these topics would be sufficient in this massive article.

Italics should be used for all Latin names, and a consistent way of writing names (eg. Aspergilli which is rather Aspergillus spp.) sometimes it is written with a small letter. Please use either the scientific names or the common ones like ascomycetes and basidiomycetes with a small letter. 

I did not find a link to bacteria however they are a key player in lichens and Glomeromycota biology.

Overall, after modifying the illustrations in the manuscript I recommend its prompt publication as a valuable guide to fungal secretion.

Minor:

Paragraph 4.3 has several issues resulting from corrections with too many options for saying the same thing in one line (eg. facilitate nutrient uptake of nutrients. Generally.therefore) l.444-446.

Zygomycetes were abandoned, for a most recent phylogenomics reference see Li, Yuanning, et al. "A genome-scale phylogeny of the kingdom Fungi." Current Biology 31.8 (2021): 1653-1665.  (line 855).

Spacing issues L.126, l.240 L.256 l.664 l.668 l.894

Missing parenthesis l.889

Author Response

All authors extend their sincere thanks for your corrections and comments, which have improved the quality of the re-submitted manuscript.

The manuscript “Turning inside out: filamentous fungal secretion and its applications in biotechnology, agriculture and the clinic” by  Timothy C. Cairns, Xiaomei Zheng, Ping Zheng, Jibin Sun and Vera Meyer summarizes the ocean of knowledge on fungal secretion. The article reviews more than 200 papers, some of them co-authored by the authors. The manuscript structure is clear and examples of genes/protein products are well suited to describe the main mechanisms described. I find it particularly useful for inexperienced mycology adepts to get the basics and be directed to the relevant literature. 

We are very grateful for these comments.

My main concern is the graphics, both figures do not relate well with the manuscript whereas several topics would be much more informative when illustrated. Let me explain this suggestion. Instead of Fig. 1. one could show the outline of the review which is a conscious selection of topics by the authors. Fig.2. would benefit from a cellular context, the compounds could be placed within a hyphae, with dedicated transporters, Golgi structures, SPK, etc. 

We agree with this point and have included 3 additional Figures, which schematically depict secretion of protein (Figure 3), secondary metabolites (Figure 4), and organic acids (Figure 5). 

Moreover, I strongly suggest showing differences between Dikarya and non-Dikarya fungi, Saccharomycotina and Pezizomycotina in chapter 2 on canonical secretion, endocytosis and extracellular secretory vesicles. 

We agree this is an important point that was not adequately addressed in the original manuscript and we are grateful for this improvement. We therefore now state at the beginning of the manuscript in section 1.0 that:

‘We highlight various different metabolic and secretory pathways for proteins (Figure 3), secondary metabolites (Figure 4), and acids (Figure 5) and provide some key examples of their applications. Key secretion mechanisms are mainly summarized from model organisms (predominantly Ascomycota). Although there are certainly important differences in secretion machinery across the fungal kingdom (e.g., between Dikarya (Ascomycota and Basidiomycota) and non-Dikarya, between flagellated and non-flagellated fungi, or between phyla [14]) these are outside the scope of this review.’

With the new citation (suggested by the reviewer below) given as reference [14]: Kiss, Enikő, et al. "Comparative genomics reveals the origin of fungal hyphae and multicellularity." Nature communications 10.1 (2019): 1-13.

Our aim in addressing the reviewers’ point in this way is to keep the important applications of secretion from the above groups (e.g., ascomycete and mucorales), while also not leading to the incorrect conclusion that mechanisms of secretion across these groups are identical.

We have highlighted this point further in other key areas of the review, e.g., section ‘2.0 Polar Growth’:

‘Recent insights into the coupling of secretion and polar growth, mainly summarized from ascomycete model organisms, are summarized below.’

And the Legend for Figure 3:

‘Note that key differences to the secretory machinery (e.g., protein components of SPK, exocyst) exist between phyla, and are still being elucidated.’

I would find it useful to present a graphical overview of types of molecules obtained from fungi with their secretion mechanisms, transporters, and the actual organism used in the industry to produce this compound.

We agree, and have summarized exemplar information in the new Figures and also Table 1, which was suggested by reviewer number 3.

Authors may decide to focus on Dikarya alone leaving away the issue of SPK and cytoskeleton differences. Otherwise, it would be recommendable to give at least a hint that GPCR repertoire, actin, dynein and other filament types are different in flagellated fungal lineages (eg. Kiss, EnikÅ‘, et al. "Comparative genomics reveals the origin of fungal hyphae and multicellularity." Nature communications 10.1 (2019): 1-13.). 

As stated above, we have highlighted this point in section ‘2.0 Polar Growth’ and other sections:

‘Recent insights into the coupling of secretion and polar growth, mainly summarized from ascomycete model organisms, are summarized below.’

If Mucorales are to stay it is essential to underline they have a different cell wall composition (Bartnicki-Garcia, Salomon. "Cell wall chemistry, morphogenesis, and taxonomy of fungi." Annual Reviews in Microbiology 22.1 (1968)) what implies PAMP repertoire (glucan!). Their structure has some resemblance to the SPK but it does not necessarily have all the identical proteins out there.

Thank you, this is an import omission that we have clarified in section ‘2.0 Polar Growth’ This section was edited to include the above point with additional citations added:

  1. Mélida H, Sain D, Stajich JE, Bulone V. Deciphering the uniqueness of Mucoromycotina cell walls by combining biochemical and phylogenomic approaches. Environ Microbiol. 2015; doi:10.1111/1462-2920.12601
  2. Lecointe K, Cornu M, Leroy J, Coulon P, Sendid B. Polysaccharides cell wall architecture of mucorales. Frontiers in Microbiology. 2019. doi:10.3389/fmicb.2019.00469

While spore dispersal can occur across very large distances (including between continents [19]), fungi are in the vast majority of cases non-motile, and consequently filamentous species colonize substrates by extension of highly polar cells termed hyphae. The shape of the hyphae is dependent on the cell wall, which is a complex, yet spatially organized, mesh of polysaccharides and proteins (recently reviewed in [20]). Abundant sugars in the cell walls of Ascomycota are 1,3-β-glucan, 1,4-β-glucan, 1,6-β-glucan, and chitin, the latter of which is composed of N-acetylglucosamine monomeric units [21]. Additional sugars include galactomannan, galactosamine, glycosaminoglycans, alpha-1,3 glucans, and others. Although cell wall composition is drastically different in other phyla (e.g., Mucorales have lower levels of 1,3-glucan and higher chitin content compared to ascomycete N. crassa [22,23]), the function of the cell wall remains comparable, with cell shape and substrate colonisation largely dependent on this structure. Consequently, the various life-styles and applications of filamentous growth is dependent on the cell wall, including branching mycelial networks in terrestrial and marine habitats [24], physical penetration of the host tissue and immune recognition during disease (section 5), or development of complex macromorphological structures during submerged fermentation (this section). Secretion plays a crucial role in establishing and maintaining filamentous morphology by delivery of membrane and cell wall synthesizing enzymes to the apex.

It would be great if authors could provide at least a suggestion of lipid and alkaloids secretion examples and applications/significance. I found only carotenoids and trichosporic acid as the terpenoid examples and no lipids, for instance PUFA. One or two sentences summarizing these topics would be sufficient in this massive article.

We have introduced PUFA secretion in section ‘6.0 Infection’:

‘More recently, the role of secreted polyunsaturated fatty acids (PUFAs) during infection has gained attention in mediating both plant and mammalian infection (reviewed in [132]).’

With reference 132:

‘Fischer GJ, Keller NP. Production of cross-kingdom oxylipins by pathogenic fungi: An update on their role in development and pathogenicity. Journal of Microbiology. 2016. doi:10.1007/s12275-016-5620-z’

Alkaloids were introduced in section 3.3, thus:

‘Marine fungi also serve as similar reservoirs for novel secondary metabolite discovery, such as bioactive alkaloids (reviewed in [99]).’

‘Willems T, De Mol ML, De Bruycker A, De Maeseneire SL, Soetaert WK. Alkaloids from marine fungi: Promising antimicrobials. Antibiotics. 2020. doi:10.3390/antibiotics9060340’

Italics should be used for all Latin names, and a consistent way of writing names (eg. Aspergilli which is rather Aspergillus spp.) sometimes it is written with a small letter. Please use either the scientific names or the common ones like ascomycetes and basidiomycetes with a small letter. 

Thanks for this very helpful comment- we have made these changes.

I did not find a link to bacteria however they are a key player in lichens and Glomeromycota biology.

This information was included in the original manuscript in section ‘7.0 Interspecies Communication’:

The most notable examples are arguably lichens, a composite organism of algae or cyanobacteria (photobiont) which live among a filamentous fungus (mycobiont), and, additionally, arbuscular mycorrhiza, a root symbiosis between Glomeromycete fungi and approximately two thirds of terrestrial plants.

Overall, after modifying the illustrations in the manuscript I recommend its prompt publication as a valuable guide to fungal secretion.

Minor:

Paragraph 4.3 has several issues resulting from corrections with too many options for saying the same thing in one line (eg. facilitate nutrient uptake of nutrients. Generally.therefore) l.444-446.

Thank you- this paragraph has been checked and words deleted/changed/sentences edited for better reading. We have also done this generally throughout the document.

Zygomycetes were abandoned, for a most recent phylogenomics reference see Li, Yuanning, et al. "A genome-scale phylogeny of the kingdom Fungi." Current Biology 31.8 (2021): 1653-1665.  (line 855).

Many thanks- ‘zygomycetes’ has been removed from this line and the rest of the document checked that it is not used.

Spacing issues L.126, l.240 L.256 l.664 l.668 l.894

We have checked the document for spacing issues and addressed them- thanks a lot

Missing parenthesis l.889

Added

Reviewer 2 Report

I would like to congratulate the authors for writing such a nice paper. I love it and do not have much to say.

In their review, authors consider filamentous fungal secretion across multiple disciplinary boundaries (e.g., white, green and red biotechnology  NOT BLUE?) and product classes (protein, organic acid, and secondary metabolite). They summarize the mechanistic understanding for how various molecules are secreted, and give numerous applications for extracellular products.

This paper will reach, in my opinion, a large audience and will be highly cited in the future.

Author Response

We are sincerely grateful for these positive and encouraging comments about the manuscript.

We do touch slightly on blue biotechnology (e.g., in enzyme discovery from marine organisms) but didn’t think the focus was strong enough to add into the title.

Reviewer 3 Report

Manuscript entitled „Turning inside out: filamentous fungal secretion and its applications in biotechnology, agriculture and the clinic” presents a broad look at the phenomenon of secretion of many different molecules by selected species of filamentous fungi. Authors present a general overview of the different applications of secreted molecules and briefly discuss the mechanisms of their secretion. The review also provides many details on specific secreted molecules used in biotechnology and interestingly describes the role of fungal secretion in interactions between organisms.

However, in my opinion the manuscript needs some improvements. One of the major suggestion is the need to organize information about the mechanisms of secretion, therefore I strongly suggest that instead of Figure 1 and Figure 2, which have already been used and published in other authors' papers, in this work, different diagrams may be prepared - including figure summarizing various mechanisms of secretion in filamentous fungi, with examples of molecules that are secreted outside the cell by different routes. The secretory mechanisms related to atypical secretion pathways, e.g. the release of various types of vesicles or biotrophic interphase complex should be discussed or illustrated in more detailed way.

I also suggest including a table that will help to organize information on industrial/biotechnological applications of individual groups of secreted molecules, along with their specific examples.

In section 6.3.4 Authors should clarify the role of Sap proteases in Candida virulence, as they are primarily involved in tissue damage and their involvement in adhesion is not directly documented.

The text should be checked for typing errors, because there are many unnecessary or missing spaces (e.g. lines 637, 665, 668 ...).

There are no authors’ names listed in the reference 78.

Author Response

All authors extend their sincere thanks for your corrections and comments, which have improved the quality of the re-submitted manuscript.

Manuscript entitled „Turning inside out: filamentous fungal secretion and its applications in biotechnology, agriculture and the clinic” presents a broad look at the phenomenon of secretion of many different molecules by selected species of filamentous fungi. Authors present a general overview of the different applications of secreted molecules and briefly discuss the mechanisms of their secretion. The review also provides many details on specific secreted molecules used in biotechnology and interestingly describes the role of fungal secretion in interactions between organisms.

However, in my opinion the manuscript needs some improvements. One of the major suggestion is the need to organize information about the mechanisms of secretion, therefore I strongly suggest that instead of Figure 1 and Figure 2, which have already been used and published in other authors' papers, in this work, different diagrams may be prepared - including figure summarizing various mechanisms of secretion in filamentous fungi, with examples of molecules that are secreted outside the cell by different routes. The secretory mechanisms related to atypical secretion pathways, e.g. the release of various types of vesicles or biotrophic interphase complex should be discussed or illustrated in more detailed way.

We agree with this point and have included 3 additional Figures, which schematically depict secretion of protein (Figure 3), secondary metabolites (Figure 4), and organic acids (Figure 5). Information above is also addressed in Table 1 (see below). 

I also suggest including a table that will help to organize information on industrial/biotechnological applications of individual groups of secreted molecules, along with their specific examples.

Thanks for this suggestion, this information is now included as Table 1.

In section 6.3.4 Authors should clarify the role of Sap proteases in Candida virulence, as they are primarily involved in tissue damage and their involvement in adhesion is not directly documented.

This sentence now reads-

Candida Saps cause disruption to host tissue [185] and degradation of the host complement system [146].’

The text should be checked for typing errors, because there are many unnecessary or missing spaces (e.g. lines 637, 665, 668 ...).

We have checked the document for all missing and double spaces.

There are no authors’ names listed in the reference 78.

Thanks, these have been added

Round 2

Reviewer 3 Report

Authors made most of the recommended changes, the work is substantive, meticulous, accurate and will certainly help researchers dealing with secretion in fungi, however it still maintain figures that have already been published elsewhere - please also indicate the license in other figures. I recommend reading the manuscript thoroughly and correct any minor typing errors. 

Author Response

Many thanks for your encouraging comments.

Figure 1 is reproduced under CC BY 4.0 as indicated in the figure legend.

Figures 2, 3, 4, 5 have been adapted from the indicated references in the Figure legend. We will confirm that these references and the 'adapted from' statements are acceptable with the Journal of Fungi Assistant Editor. We will also correct any additional typing errors e.g. at the proofing stage should the manuscript be accepted.